# Identification of Immune Function-Related Subtypes in Cutaneous Melanoma

**DOI:** 10.3390/life11090925

**Published:** 2021-09-06

**Authors:** Lin Liu, Junkai Zhu, Tong Jin, Mengjia Huang, Yi Chen, Li Xu, Wenxuan Chen, Bo Jiang, Fangrong Yan

**Affiliations:** Research Center of Biostatistics and Computational Pharmacy, China Pharmaceutical University, Nanjing 210009, China; linliu.cpu@foxmail.com (L.L.); zhujunkai97@126.com (J.Z.); tjin.cpu@foxmail.com (T.J.); chung_hmj@163.com (M.H.); yichencheny@163.com (Y.C.); 3320051716@stu.cpu.edu.cn (L.X.); chengwenxuan1997@outlook.com (W.C.)

**Keywords:** cutaneous melanoma, therapeutic targets, immune function-related subtype, immunotherapy

## Abstract

Tumour immunotherapy combined with molecular typing is a new therapy to help select patients. However, molecular typing algorithms related to tumour immune function have not been thoroughly explored. We herein proposed a single sample immune signature network (SING) method to identify new immune function-related subtypes of cutaneous melanoma of the skin. A sample-specific network and tumour microenvironment were constructed based on the immune annotation of cutaneous melanoma samples. Then, the differences and heterogeneity of immune function among different subtypes were analysed and verified. A total of 327 cases of cutaneous melanoma were divided into normal and immune classes; the immune class had more immune enrichment characteristics. After further subdividing the 327 cases into three immune-related subtypes, the degree of immune enrichment in the “high immune subtype” was greater than that in other subtypes. Similar results were validated in both tumour samples and cell lines. Sample-specific networks and the tumour microenvironment based on immune annotation contribute to the mining of cutaneous melanoma immune function-related subtypes. Mutations in *B2M* and *PTEN* are considered potential therapeutic targets that can improve the immune response. Patients with a high immune subtype can generally obtain a better immune prognosis effect, and the prognosis may be improved when combined with TGF-β inhibitors.

## 1. Introduction

Cancer is not a single disease but a collection of multiple biological entities, and each has its own molecular characteristics and clinical significance [1]. Traditionally, cancer classification was based on histopathology and clinical characteristics, most of which depend on clinicians’ judgement. However, studies show that prognosis and treatment responses vary between cancer subtypes and within subtypes. Therefore, it is necessary to develop methods that can accurately predict the key results of individual patients [2,3]. Although some progress has been made in the field, there are still unexplained inter-tumoural heterogeneities leading to varying survival outcomes and differences in treatment responses [4]. Gene expression features are easy to measure from tissue samples, and may be used to identify personalized “driving mutations” [5], differentially expressed genes and pathways [6], and personalized gene networks [7]. Although genes have been successfully used as biomarkers for cancer diagnosis [8], it is not clear whether gene biomarkers are the most suitable method for treatment indicator classification. Conversely, it may be more meaningful to describe diseases by using system-specific dysfunction rather than the dysfunction [9,10] of individual molecules, which is based on the gene regulatory network.

Currently, intergenic relationships can be obtained in three ways: (1) gene regulation networks based on existing public database information; (2) gene regulation network based on intergenic correlation [11] or causality inferred [12]; and (3) gene regulation network based on an annotated database by prior information combined with gene expression data [13], which is called a mixed method. Most of the existing studies build networks based on information from all samples. In 2007, Borgwardt et al. [14] proposed using the PPI network as a reference network template, and the edges were tailored according to the gene co-expression state of the individual relative to the whole population. In 2009, Kuijjer et al. [15] further proposed the LIONESS method, which constructs personalized networks by evaluating the contribution of each sample to gene regulatory networks. The two methods have a common disadvantage: when a new sample enters the network, all network information needs to be recalculated. In 2019, Thin Nguyen et al. [16] proposed a new method to construct a sample-specific network called the “personalized annotation-based networks” (PAN). This method uses an annotated database information to transform gene expression data into a network, where nodes correspond to functional terms and edges correspond to the correlation between points. Since PAN considers a single sample that discards common information between samples, it can effectively avoid the defects of recalculation.

As a representative immunotherapy, cutaneous melanoma (SKCM) has a high mortality rate given its high malignancy, high incidence, and easy metastasis [17]. At present, immunotherapy is effective in the treatment of skin melanoma, but its disadvantages, such as low response rate, significant differences in immune response in different patients and unexpected adverse reaction occurrence, make it less effective [18,19]. Therefore, how to select patients suitable for immunotherapy is an urgent issue [20]. As molecular typing based on immune function can identify the specific biological characteristics of the population to adapt to specific treatments [21], it is an effective means to improve the efficacy of SKCM immunotherapy. However, the relationship between SKCM subtypes and immune function has not been well explored.

Here, we comprehensively analysed the molecular profiles of SKCM, including omics characteristics, immune mechanisms, and stromal cell infiltration abundance based on sample-specific networks, to identify subtypes related to immune function in SKCM and the molecular characteristics behind their subtypes.

## 2. Materials and Methods

### 2.1. Patients and Samples

The discovery cohort used in this study was from The Cancer Genome Atlas (TCGA) under the SKCM archive with a total of 327 tumour samples; gene expression, mutation, DNA methylation, and clinical data were retrieved. An external validation cohort including 214 SKCM data and paracancerous tissues was obtained from the GEO database (GSE7553 [22] [*n* = 82], GSE23376 [23] [*n* = 22], GSE19234 [24] [*n* = 44], GSE15605 [25] [*n* = 58]), and 206 tumour samples (Appendix A) were merged based on elimination of batch effects. Immune annotation information was obtained from the MSigDB database. The data of cell lines [26] (*n* = 49) in cutaneous melanoma were downloaded from DepMap (https://depmap.org/portal/, accessed on 9 August 2021).

### 2.2. Sample-Specific Networks Were Constructed with the SING Method

We selected the C7 data set in the MSigDB database as the annotated database and the modified PAN method as SING (single sample immune signature network), which followed the analytic process below (Figure 1).

Feature selection

(1)The high-dimensional nature of gene expression data makes it necessary to reduce dimensionality before analysis. For each gene data set, the top 500 genes were selected according to the order of their total variance and included in subsequent analysis and calculation.(2)Construction of gene and immune annotation correlation matrix(3)$M_{p,q}^{j}$ is the correlation matrix between the gene and immune annotation, in which j corresponds to the j sample, p corresponds to the first gene, and q corresponds to the first annotation information in the immune annotation database. The corresponding element in the matrix (line p, column q) is the expression value of p; otherwise, it is recorded as 0.(4)Development of sample-specific network(5)The symmetric correlation matrix $A_{p,q}^{j}$ is further generated by the matrix $M_{p,q}^{j}$. Each element in the matrix corresponds to the Euclidean distance between two annotation pieces of information q and $q^{*}$. Afterward, to transform into a discrete network, this study further selected the edge of the top 10% weight in the symmetric correlation matrix $A_{p,q}^{j}$ to construct the adjacency matrix.(6)Computing network topology information.

Once the discrete network construction of each sample was completed, the topological property “degree” of the network was calculated and obtained for subsequent clustering analysis. For a given sample S, its network is represented as $G^{S}\left(V^{S},E^{S}\right)$, and the corresponding topological property is denoted as $h^{s}$, which is a vector, $h^{s}=\left[h_{1}^{s},\cdots ,h_{\left|V^{s}\right|}^{s}\right]$
$$E\left(v_{j}^{s},v_{k}^{s}\right)=\{\begin{array}{c} 1\;,\;\\ 0\;, \end{array}\;\;\;\;\;\;\;\;\;\;\begin{array}{c} if\;j\;is\;connected\;with\;k\\ otherwise \end{array}\;$$
$$h_{j}^{s}=\sum_{v_{k}^{S}\in V^{S}-\left\{V_{j}^{S}\right\}}l\left(v_{j}^{s},v_{k}^{s}\right)$$

### 2.3. Clustering Algorithm

Clustering is an effective algorithm for molecular typing and has been successfully applied in the molecular typing of many kinds of cancer [1,27]. We used a model-based clustering algorithm based on the Gaussian finite mixture model (R package “mclust” v5.4.7) to find the best cluster numbers of the SKCM subtypes. For cancer subtype classification, we used hierarchical clustering and K-means algorithms. In the validation cohort, supervised consensus clustering was used to verify the existence of immune subtypes. The gene expression profile was resampled 500 times. In each resampling, 80% of the samples and 80% of the genes in the original matrix were taken. During each clustering process, the distance was measured by the 1-Pearson correlation coefficient, and the linkage was set as error sum of square (ESS) between classes.

### 2.4. Assessment of Immune and Stromal Cell Infiltration

As the degree of infiltration is related to the prognosis of immunotherapy in immune and stromal cells, we used the R package “MCPcounter” (v1.2.0) [28] to calculate the abundance fraction of eight immune cells (T cells, CD8 + T cells, NK cells, cytotoxic lymphocytes, B cell lineages, single-cell lineage cells, myeloid dendritic cells, and neutrophils) in the tumour microenvironment and two stromal cells (endothelial cells and fibroblasts). Then, “xCell” (v1.1) [29] was used to calculate 64 immune and nonimmune cell types, and “ESTIMATE” (v1.0.13) [30] was used to calculate the immune cell and stromal cell infiltration score.

### 2.5. Cytolytic Activity

Cell dissolution activity, also known as cell killing activity, can be used to characterize the immunogenic nature [31] of tumours. We obtained the cytolytic activity by calculating the geometric mean of granzyme A (GZMA) and perforin 1 (PRF1) gene expression in each sample.

### 2.6. Gene Differential Expression Analysis

The standard analysis process of “DESeq” (v1.30.1) [32] was used to analyse the differential gene expression of the original count data obtained by RNA sequencing. Nominal *p* values were adjusted by the false discovery rate (FDR). Genes with fold change ≥ 2 and FDR < 0.05 were considered differentially upregulated genes, while those with fold change ≤ 1/2 and FDR < 0.05 were considered differentially downregulated genes.

### 2.7. Differential Gene Enrichment Analysis

The gene enrichment analysis included GO functional annotation and KEGG pathway analysis. The GO functional annotation was completed by the DAVID 6.8 [33,34] online database, and KEGG pathway analysis was performed by the R package “clusterProfiler” (v3.18.1) [35]. Single-sample gene set enrichment analysis (ssGSEA) was performed based on the R package “GSVA” (v1.38.2) [36] to obtain the enrichment fraction of the gene set in a single sample. The scores of 24 microenvironment cell types were calculated, and the final microenvironment cell abundance was further adjusted by tumour purity through the formula score/(1− tumor purity).

### 2.8. Recognition of Cancer Mutation-Driven Genes and Calculation of Tumour Mutation Load (TMB)

Based on cancer somatic mutation data, we used the R package “MutSigCV” (v2.0) [37] to identify the significantly driving mutated genes of the subtypes (q < 0.05). Then, “maftools” (v2.6.05) were used to calculate the TMB in each subtype [38] to reflect mutations in tumour cells.

### 2.9. DNA Methylation Differential Analysis

We analysed two methylation statuses of the promoter: (1) hypermethylation status and (2) demethylation status. First, low-quality probes were filtered and differentiated CpG sites were identified. This result suggested that when log2FoldChange > 0 and the adjusted *p* value < 0.01, the probe showed a hypermethylation level; when log2FoldChange < 0 and the adjusted *p* value < 0.01, the probe showed a low methylation level (R package “ChAMP” v2.20.1) [39].

### 2.10. Survival Analyses

To confirm whether there was a significant difference in survival between subtypes, a Kaplan–Meier curve [40] was used for survival analysis and the log-rank test for detection of significant difference (*p* < 0.05). Univariate Cox proportional hazards regression was used to calculate the predictors associated with cancer survival [41]. After integrating all predictors, the nomogram [42] was used to reflect the correlation between the variables in the prediction model. A multivariate Cox regression model according to the published literatures [43,44] was used to calculate the risk score, which showed prognostic value.

## 3. Results

### 3.1. Determination Aand Verification of Cluster Number

Based on the degree topological property matrix of immune annotation sample-specific as input matrix, the optimal number of clusters were tested from one to six. The sample data were divided into two categories (Figure 2A). Then, the hierarchical clustering algorithm where the distance was measured by 1-Pearson’s coefficient and the clustering method was set as ward. Finally, 146 samples fell into the first category, called the C1 class and 181 samples fell into the second category, called the C2 class.

To verify the validity of the clustering results, the two clustered categories were compared with the TCGA classification results. A high correlation was found between them, and the immune subclass mainly overlapped with the C2 class (Figure 2B). Furthermore, the two categories were compared with another published dataset (Figure 2C) that contained 47 patients with SKCM [45] who responded to immunotherapy. C2 may have a high likelihood of responding to anti-PD-1 therapy (Bonferroni corrected *p* = 0.008). Then, univariate Cox regression was used to analyse those factors that may affect the survival of SKCM patients (Appendix A). Age at diagnosis, body weight, race, tumour stage, UV markers, UV rate, metastatic status, and ploidy were significantly associated with patient prognostic survival (*p* < 0.05). The nomogram (Figure 2D) can predict the survival rate of patients at different factor levels.

### 3.2. Definition and Extraction of Immune Subtypes

To explore the immune characteristics in SKCM, we defined subtypes of C1 and C2 based on the immune and matrix score, infiltration abundance of immune and stromal cells, and cytopathic activity. As the C2 class was more immune-infiltrated, we defined the C1 class as the “normal class” and C2 as the “immune class”.

The optimal clustering number of the immune class was three according to the BIC value (Figure 3A). Based on the K-means clustering method, the number of samples for the three classes were 38, 133, and 10. Combining immune clustering with normal class (Appendix A), “immune2 class” can be significantly separated from other classes. “Immune3 class” and “normal class” had high similarities in the tumour microenvironment. In addition, there was no significant difference in the degree of immune infiltration between the “normal class” and “immune3 class” (Appendix A). The immune infiltration of the “immune1 class” and “immune2 class” were significantly different. Therefore, the “immune3 class” and “normal class” were combined as the “new-normal class” in this study. Comparing the tumour microenvironment and immune score of the three types (Figure 3B,C), we found significant differences among them. “xCell” was used to calculate the invasion abundance of tumour immune and stromal cells. Tumour immune infiltration in the “immune 2 class” was significantly higher than that in the other two subtypes (Figure 3D). A significant prognostic difference among them (*p* = 0.0039) was also observed: the “new-normal class” was the worst, followed by “immune1 class”, and “immune2 class” was the best (Figure 3E).

Combined with the immune score and survival prognosis of each subtype, we renamed the “new-normal class” the “immune inactivation subtype”, the “immune 1 class” the “low immune subtype” and the “immune 2 class” the “high immune subtype”.

### 3.3. Heterogeneity Analysis between High and Low Immune Subtypes

Based on differential gene expression analysis, the expression of 1030 genes in the “high immune subtype” were significantly upregulated relative to that in the “low immune subtype”, and 87 genes were significantly downregulated (Figure 4A). Compared with the immune cell gene set, 240 genes were found to be immune-related among the significantly upregulated genes in the “high immune subtype”, while only one immune-related gene was significantly downregulated. These results indicate that the “high immune subtype” had more immune-enriched characteristics.

Based on 1117 differentially expressed genes between the “high immune subtype” and “low immune subtype”, GO functional annotation and KEGG pathway enrichment were further analysed. These differentially expressed genes were mainly related to the immune response, inflammatory response, T cell proliferation, leukocyte proliferation, and chemokine-mediated signalling pathways (Figure 4B). Using KEGG pathway analysis, 16 signalling pathways were identified (Appendix A). Among those, 13 also appeared in the normal and immune classes. The PI3K-Akt, MAPK, and TGF-β signalling pathways appeared in the enrichment pathway of differentially expressed genes between the “low immune subtype” and “high immune subtype”. Several KEGG pathways were also found to be related to Th1 cell differentiation, Th2 cell differentiation, Th17 cell differentiation, and PD-L1 expression. They showed an increasing trend among the “immune inactivation subtype”, “low immune subtype” and “high immune subtype” (Appendix A). There were significant differences (Th1: *p* < 2.2 × 10^−16^, Th2: *p* = 3.5 × 10^−11^, Th17: *p* = 4.0 × 10^−12^), and the abundance of MDSCs was the highest in the “immune inactivation subtypes” and lowest in the “high immune subtype” (*p* = 9.1 × 10^−15^).

The influence and interaction between various factors in the tumour were complex, and the methylation status and gene mutations in the tumour had a strong relationship with gene expression, which would affect tumour occurrence and development. Therefore, we continued to explore the differences between the “high immune subtype” and “low immune subtype” based on tumour methylation and gene mutation. In this study, 21,981 probes were screened for methylation differences between high and low immune subtypes; 982 probes were hypermethylated, and 4620 probes were demethylated. After mapping these probes to genes and annotating with immune gene sets, a total of 597 genes showed hypermethylation but were epigenetically silenced in the “high immune subtype”, among which 10 genes were immune system related. In addition, 1490 genes were demethylated, of which 60 were related to immunity. Then, six significantly mutated genes were identified with a *p*-value < 0.01, including *NRAS*, *BRAF*, *B2M*, *PTEN*, *TP53*, and *CDKN2A* (Appendix A). NRAS appeared in both the “high immune subtype” and the “low immune subtype”. In addition, *BRAF* and *CDKN2A*, which are star mutations in SKCM [46], appeared in the “high immune subtype”, and the mutation frequency of BRAF in the “high immune subtype” was greatest. *BRAF* is commonly seen in the MAPK pathway (RAS/RAF/MEK/ERK pathway), and mutation of this pathway is an important factor that causes growth and development of melanoma. These results indicate that *NRAS* and *BRAF* mutations were specific characteristics of the two subtypes. Additionally, *B2M* and *PTEN* mutations occurred only in the “immune inactivation subtype”.

There were no significant differences in TMB levels between the “high immune subtype” and “low immune subtype”. Because the TMB level of the “immune class” was significantly higher than that of the “normal class” (*p* = 0.00058; Appendix A), we suggested that the immune class was more likely to benefit from immunotherapy. Moreover, the “high immune subtype” was more promising for PD-1 treatment [45] (Bonferroni correction *p* = 0.03; Figure 4D). Cell cytotoxicity and cytolysis are also markers of tumour immunogenicity. The cytotoxicity and cytolysis of the “high immune subtype” was significantly higher than that of the “low immune subtype” (*p* = 3.2 × 10^−9^; Figure 4C).

### 3.4. Immunity Subtype Validation

In this study, four independent cohorts were integrated as a verification set for supervised analysis (206 tumour samples). First, the existence of an immune class (Cluster 1) and a normal class (Cluster 2) were verified, with the immune class having higher immune signatures (Appendix A). To further validate immune subtypes based on the tumour microenvironment, three subtypes of the TCGA cohort were analysed for pairwise differential expression, and the corresponding differentially expressed genes were selected as the specific signatures of the three SKCM immune subtypes. Among them, a 737-gene signature was identified for “immune inactivation subtypes”, a 525-gene signature for “low immune subtype”, and an 839-gene signature for “high immune subtype”. Using the above characterization gene signatures for supervised consensus clustering (Figure 5A), three subtypes of the validation cohort were reproduced: “high immune subtype” (98 samples), “low immune subtype” (58 samples), and “immune inactivation subtypes” (50 samples). Among the three subtypes, significant differences were observed in cell cytotoxicity and cytolysis, immune score, and stromal score (Figure 5B–D, *p* < 0.05).

Then, we validated the results on cell lines of cutaneous melanoma. We applied the SING method on the cell lines data (*n* = 49), the optimal clustering number was three according to the BIC value (Appendix A). Comparing the tumour microenvironment, stromal score, and immune score of the three clusters (Figure 5E–G, *p* < 0.05), we found significant differences among them. Tumour immune infiltration in the “C1” was significantly higher than that in the other two clusters, which was highly similar to the “high immune subtype”. However, cytolytic activity score among three clusters were not significant as the expressions of the cell lines were too low (Appendix A). Above all, these results suggested that specific immune subtypes were present in cutaneous melanoma and that their immune responses varied according to subtype characteristics in tumour samples and cell lines. 

### 3.5. Specific Pathological and Prognostic Features of Immune Subtypes

For 327 samples, the heterogeneity in clinical features among the three immune subtypes was explored from the following six perspectives (Table 1): (1) age, (2) tumour registration, (3) race, (4) body mass index (BMI), (5) metastatic status, and (6) UV exposure rate. We did not observe differences in age, race, BMI, or UV exposure rates among subtypes, but significant differences were found in tumour grade and tumour status (*p* < 0.05). Specifically, more tumour grade samples were III or IV in the “high immune subtype” (“immune inactivation subtype” vs. “low immune subtype” vs. “high immune subtype”: 37% vs. 37% vs. 50%, *p* = 0.05), and the proportion of metastatic samples was higher (“immune inactivation subtype” vs. “low immune subtype” vs. “high immune subtype”: 69% vs. 87% vs. 94%, *p* = 6.21 × 10^−8^).

We calculated the risk score using a multivariate Cox regression model. Based on the eight-immune-related genes (IRGs) signature (*PSME*, *CDC42*, *CMTM6*, *HLA-DQB1*, *HLA-C*, *CXCR6*, *CD8B*, *TNFSF13*) and the five-immune-associated gene (IAG) signature (*IFITM1*, *TNFSF13B*, *APOBEC3G*, *CCL8*, and *KLRK1*), both published signatures [43,44] showed prognostic value when reproducing them in our study (*p* < 0.001, HR = 1.765, 95% CI: 1.297–2.402; *p* = 0.0667, HR = 1.328, 95% CI: 0.9807–1.797). Additionally, we found significant differences in the risk score between the three immune subtypes (*p* < 0.05). Among them (Appendix A), “high immune subtype” had a lower risk score, suggesting that “high immune subtype” may have a better prognosis, lower tumour purity and active immune-related signalling pathways.

## 4. Discussion

In recent years, research on tumour immunotherapy has made rapid progress, and immunotherapy led by SKCM has received extensive attention [18]. At present, molecular typing based on immune function is considered an effective method to improve the efficacy of tumour immunotherapy [47] because it can identify people with specific biological characteristics for valuable treatment. The construction of a sample gene network [48] can help to characterize complex and heterogeneous multigroup data sets and improve the accuracy of subtype identification and subsequent analysis. LIONESS and PAN [14,15,16] have already been proven to identify subgroups of different disease types and to solve clinically related classification problems.

We herein started by considering the tumour microenvironment and tumour cells and built the SING method by immune annotation information as background to explore the immune function of cutaneous melanoma-related subtypes of skin. The SING algorithm is helpful to identify immune functions related to subtypes. The “immune inactivation subtype” had two unique genetic mutations, *B2M* and *PTEN*, which meant that they were molecular features of the normal class. Studies have found that B2M is associated with antigen presentation in the tumour immune cycle [49]. *B2M* gene-encoded β-20 microglobulin (MHC-1) is an important adjunct to major histocompatibility complex class 1, and its absence leads to a decline in MHC-1 expression, affecting antigen presentation and resulting in PD-1 drug resistance [50]. A meta-analysis to systematically review [51] showed that PD-1 inhibitors significantly improved the progression-free survival (PFS), overall survival (OS) and overall response rate (ORR) in patients with advanced melanoma. Anti-PD-1 drugs such as pembrolizumab and nivolumab can achieve long-term survival for patients with metastatic melanoma [52]. *PTEN*, as a tumour suppressor gene [53], was found only in the normal class, suggesting that PTEN mutations may be a factor affecting immune properties. Therefore, we suggest that they are potential therapeutic targets to improve the immune response. Compared with the “low immune subtype”, “high immune subtype” showed higher immune infiltration and immune activation characteristics and was significantly enriched in TGF-β signalling pathways [54,55]. TGF-β signalling pathways have been repeatedly reported [54,56,57] to be associated with immune regulation, and the use of TGF-β inhibitors may improve the immune response to immunotherapy. In recent years, inhibitors targeting TGF-β pathway have been discovered and investigated by pharmaceutical companies for cancer therapy, and some of them are in clinical trial now, such as Phase I study of GC1008 (fresolimumab), Ph II study of NCT01453361 (Gemogenovatucel-T) and so on [58,59]. In 2019, Kaczorowski et al. [60] suggested that overexpression of SMAD7 may be a new hallmark inhibitor of TGF-β in melanoma. In 2021, Liu et al. [61] proposed a triple combination therapy with PD-1/PD-L1, *BRAF*, and *MEK* inhibitor in stage III-IV melanoma as significantly improving PFS of patients. Based on the risk score between the three immune subtypes, “high immune subtype” showed better prognostic value. Therefore, we speculated that patients with “high immune subtype” biological characteristics can generally obtain a better immune prognosis and that the immune prognosis is improved if combined with TGF-β inhibitors. In the “low immune subtype”, only one significantly mutated gene, *NRAS*, was found, which was considered a molecular feature of this subtype.

Although we divided the immune subtypes of cutaneous melanoma from the effects of tumour cells and the tumour microenvironment, there were still some inextricable links among various factors that we did not consider. We also have not considered the interaction between factors from comprehensive perspectives. Not with-standing its limitation, this study does suggest that the research of cancer typing based on immune function can be enhanced by using a sample-specific network. Further study is needed on how to combine the key factors that influence the response to cancer and treatment to conduct a complete analysis of cancer.

## 5. Conclusions

In summary, this comprehensive study classified cutaneous melanoma skin samples based on immune function and proposed three subtypes according to SING and the tumour microenvironment. Mutations in *B2M* and *PTEN* are considered potential therapeutic targets that can improve the immune response. “High immune subtype” patients can generally obtain a better immune prognosis effect if TGF-β inhibitors are combined.

## Figures and Tables

**Figure 1 life-11-00925-f001:**
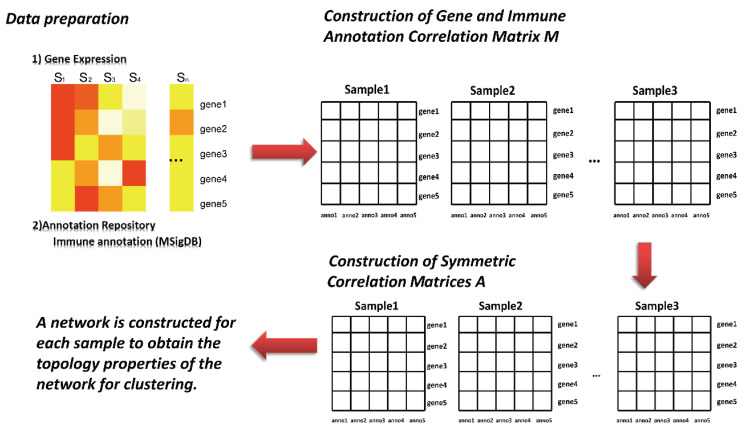
Flowchart of Sample Specific Network Algorithm. (1) Prepare data for feature selection. (2) Construct the gene and immune annotation correlation matrix. (3) Build a sample-specific network. (4) Compute network topology information.

**Figure 2 life-11-00925-f002:**
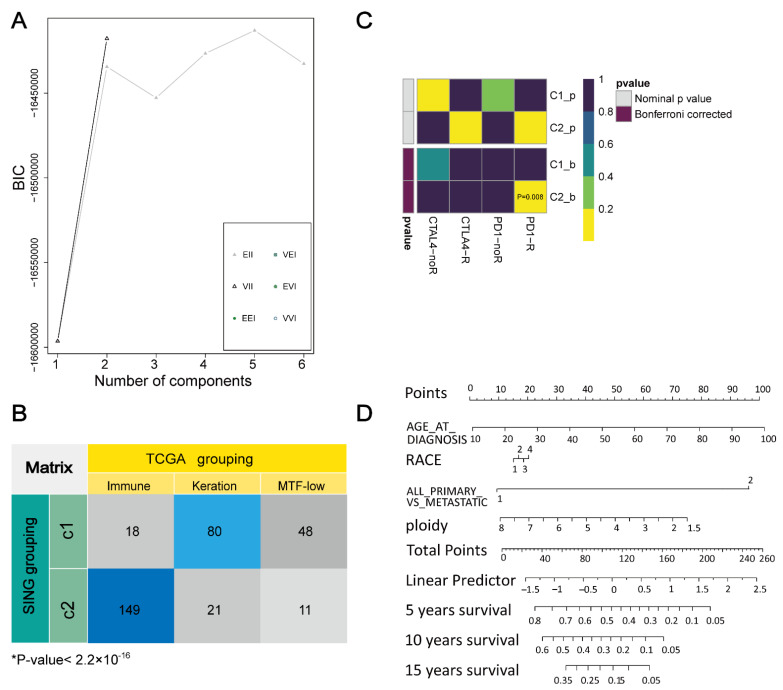
Define and validate the best SKCM cluster number. (**A**) Determination of the optimal cluster number (k = 2) according to BIC values. (**B**) Comparing the two classes and TCGA classification, the “immune subclass” mainly overlapped with the C2 class (*p* < 2.2 × 10^−16^). (**C**) Submap analysis showed that the C2 class may be more sensitive to programmed cell death protein 1 inhibitors (Bonferroni corrected *p* = 0.008). (**D**) The nomogram directly reflected the correlation between variables in the prediction model.

**Figure 3 life-11-00925-f003:**
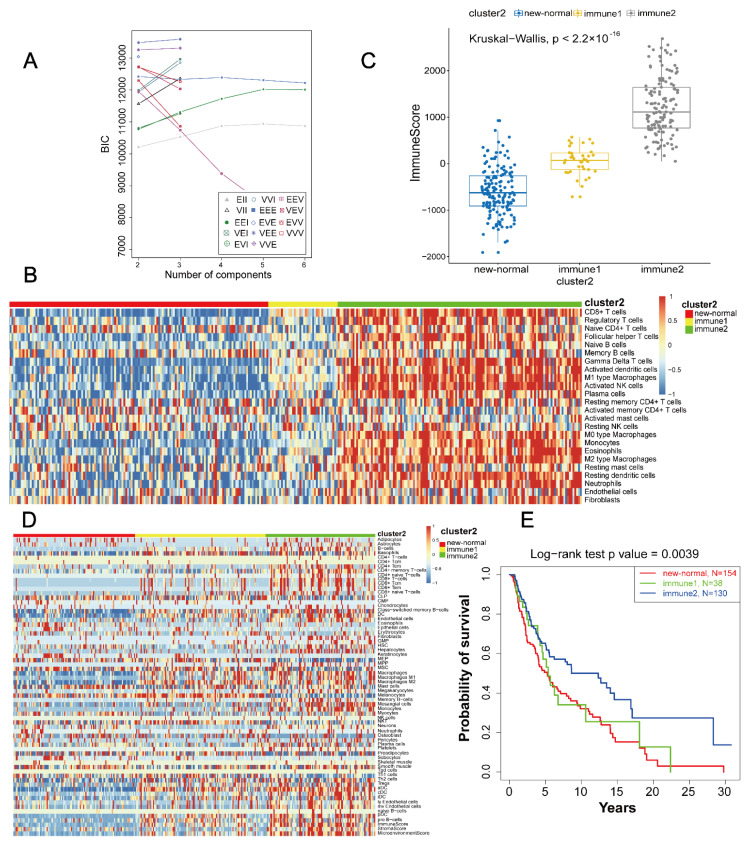
Definition and extraction of all subtypes. (**A**) Determination of the optimal clustering number in immune type according to BIC values (k = 3). (**B**) Heatmap of tumour microenvironment in three subtypes. (**C**) Significantly different immune scores among the three subtypes (*p* < 2.2 × 10^−16^). (**D**) xCell immune and stromal cell infiltration abundances in the three subtypes. (**E**) Immune 2 (“high immune subtype”) had the best survival analysis among the three subtypes (*p* = 0.0039).

**Figure 4 life-11-00925-f004:**
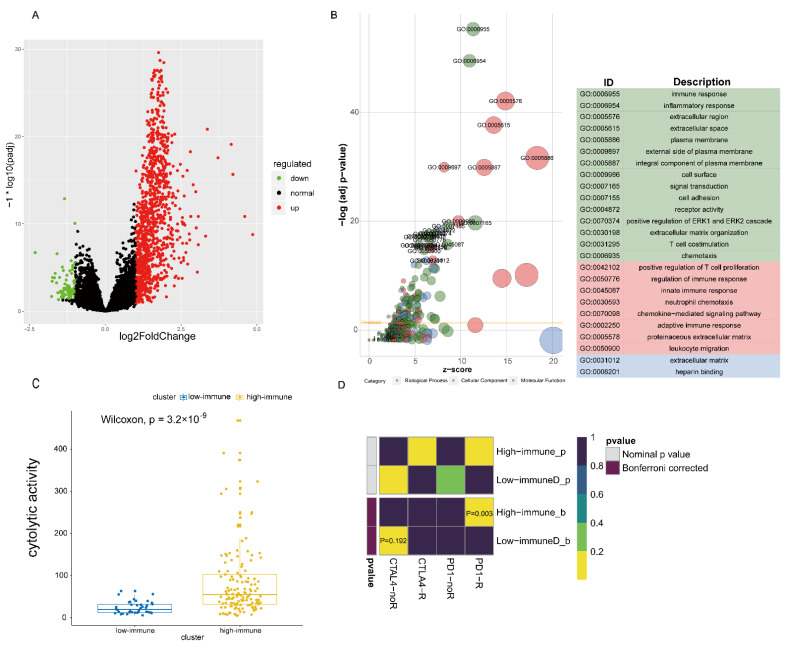
Heterogeneity analysis between high and low immune subtypes. (**A**) Volcano plot of differential gene expression in high and low immune subtypes. (**B**) GO functional annotation bubble chart for 1117 DEGs. (**C**) Comparison of cytotoxicity levels in high and low immune subtypes (*p* = 3.2 × 10^−9^). (**D**) Submap analysis suggested that the “high immune subtype” might be more sensitive to PD-1 inhibitors (Bonferroni corrected *p* = 0.03).

**Figure 5 life-11-00925-f005:**
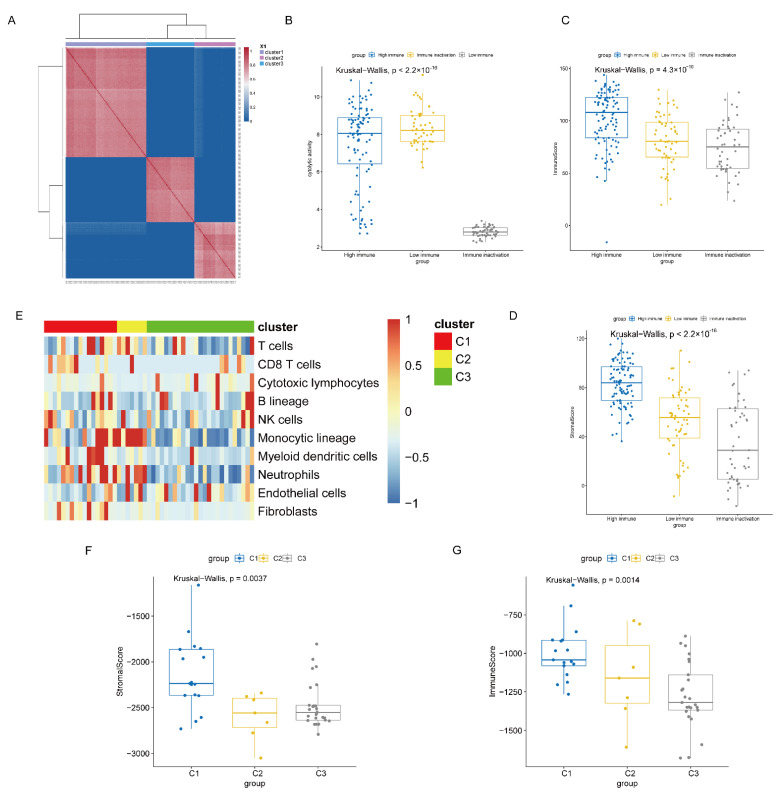
Identifications of “high immune subtype”, “low immune subtype”, and “immune inactivation subtype”. (**A**) Consensus clustering by using three gene signatures on validation sets. (**B**–**D**) Significant differences among the three subtypes were found in cell cytotoxicity and cytolysis, immune score, and stromal score (all, *p* < 0.05). (**E**–**G**) Significant differences among the three clusters in cell lines were found in tumour microenvironment, immune score, and stromal score (all, *p* < 0.05).

**Table 1 life-11-00925-t001:** Statistics of pathological characteristics of immune subtypes and differences among three subtypes.

KERRYPNX		FRE (%)	Specific Immune Subtype			*p*-Value
			Immune in Activation Subtype	Low Immune Subtype	High Immune Subtype	
Age	≤60	179 (55)	79	23	77	0.64
	>60	144 (44)	75	15	54	
	NA	4 (1)	2	0	2	
TNM Stage	0 + I + II	163 (50)	89	21	53	0.05
	III + IV	137 (42)	57	14	66	
	NA	27 (8)	10	3	14	
Race	White	318 (97)	150	37	131	0.43
	Asians	7 (2)	5	1	1	
	Black or African American	1 (0)	0	0	1	
	NA	1 (0)	1	0	0	
BMI	≤25	46 (14)	27	5	14	0.52
	>25	90 (28)	42	12	36	
	NA	191 (58)	87	21	83	
Tumour Status	Primary	62 (19)	49	5	8	6.214 × 10^−8^ *
	Metastatic	265 (81)	107	33	125	
UV Exposure Rate	≤0.8	83 (25)	47	11	25	0.07
	>0.8	244 (75)	109	27	108	

* Fisher’s exact test *p*-value < 0.05.

## Data Availability

Raw data for this study were generated at TCGA with cancer type of SKCM. The datasets used and/or analyzed during the current study are available from GEO database (GSE7553, GSE23376, GSE19234, GSE15605).

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
