# Peer review of "Identification of Immune Function-Related Subtypes in Cutaneous Melanoma"

_life, 2021, doi:10.3390/life11090925_

Round 1

Reviewer 1 Report

Authors proposed a single sample immune signature network (SING) method to identify new immune function-related subtypes of cutaneous melanoma, and authors also showed that the identification of new immune function-related subtypes of cutaneous melanoma has beneficial therapeutic implication, where patients have better prognosis when combined with TGF-β inhibitors. The proposed method is novel, and the results are of interest. I think it is generally acceptable in current format but the manuscript could be still improved a little bit if the authors can consider the following decorative suggestions:

  1. Authors referred to R packages many times, but missed the version of R packages.
  2. Authors didn't specify the references of cell lines in the studies.
  3. The sentences “Both published 299 signatures showed prognostic value when reproducing them in our study in page 12-13” are not readable. Authors need to add more information about the signatures.
  4. Figures section:
    1. The numbers in the figures should be consistent with that in the manuscript.
    2. The font is very small in some panels and it is very difficult to read the figures, such as Fig3.

Author Response

Authors proposed a single sample immune signature network (SING) method to identify new immune function-related subtypes of cutaneous melanoma, and authors also showed that the identification of new immune function-related subtypes of cutaneous melanoma has beneficial therapeutic implication, where patients have better prognosis when combined with TGF-β inhibitors. The proposed method is novel, and the results are of interest. I think it is generally acceptable in current format but the manuscript could be still improved a little bit if the authors can consider the following decorative suggestions:

  1. Authors referred to R packages many times, but missed the version of R packages.
  2. Authors didn't specify the references of cell lines in the studies.
  3. The sentences “Both published 299 signatures showed prognostic value when reproducing them in our study in page 12-13” are not readable. Authors need to add more information about the signatures.
  4. Figures section:
  • The numbers in the figures should be consistent with that in the manuscript.
  • The font is very small in some panels and it is very difficult to read the figures, such as Fig3.

Response:

Thank you for your considerate comments which is very useful for us to further improve the manuscript. Please see the following details and the attachment.

-#1.Authors referred to R packages many times, but missed the version of R packages.

Response: We have added the version of R packages in our manuscript. The detail information of all R packages referred in this study was listed below.

The version information about the R package

Package

Version

mclust

5.4.7

MCPcounter

1.2.0

xCell

1.1

ESTIMATE

1.0.13

DESeq

1.30.1

clusterProfiler

3.18.1

GSVA

1.38.2

MutSigCV

2.0

maftools

2.6.05

ChAMP

2.20.1

-#2.Authors didn't specify the references of cell lines in the studies.

Response: I have added the reference in my manuscript. Here is the reference information. Please see the attachment.

Patients and Samples

The discovery cohort used in this study was from The Cancer Genome Atlas (TCGA) under the SKCM archive with a total of 327 tumour samples; gene expression, mutation, DNA methylation, and clinical data were retrieved. An external validation cohort including 214 SKCM data and paracancerous tissues was obtained from the GEO database (GSE7553 [22] [n = 82], GSE23376 [23] [n = 22], GSE19234 [24] [n = 44], GSE15605 [25] [n = 58]), and 206 tumour samples (Supplementary Table S1) were merged based on elimination of batch effects. Immune annotation information was obtained from the MSigDB database. The data of cell lines [26](n=25) in cutaneous melanoma were downloaded from DepMap (https://depmap.org/portal/).

[26] Next-generation characterization of the Cancer Cell Line Encyclopedia

-#3.The sentences “Both published 299 signatures showed prognostic value when reproducing them in our study in page 12-13” are not readable. Authors need to add more information about the signatures.

Response:  I have put more information in my manuscript. Here is the details.

Survival analyses

To confirm whether there was a significant difference in survival between subtypes, Kaplan-Meier curve [40] was used for survival analysis and the log-rank test for detection of significant difference (P < 0.05). Univariate Cox proportional hazards regression was used to calculate the predictors associated with cancer survival [41]. After integrating all predictors, the nomogram [42] was used to reflect the correlation between the variables in the prediction model. A multivariate Cox regression model according to the published literatures ([43, 44]) was used to calculate the risk score, which showed prognostic value.

We calculated the risk score using a multivariate Cox regression model. Based on the eight-immune-related genes (IRGs) signature (PSME, CDC42, CMTM6, HLA-DQB1, HLA-C, CXCR6, CD8B, TNFSF13) and the five-immune-associated gene (IAG) signature (IFITM1, TNFSF13B, APOBEC3G, CCL8 and KLRK1), both published signatures [43, 44] showed prognostic value when reproducing them in our study (P < 0.001, HR = 1.765, 95% CI: 1.297-2.402; P =0.0667, HR =1.328, 95% CI: 0.9807-1.797). Additionally, we found significant differences in the risk score between the three immune subtypes (P < 0.05). Among them (Supplementary Figure S8), “high immune subtype” had a lower risk score, suggesting that “high immune subtype” may have a better prognosis, lower tumour purity and active immune-related signalling pathways.

-#4.Figures section:

  • The numbers in the figures should be consistent with that in the manuscript.
  • The font is very small in some panels and it is very difficult to read the figures, such as Fig3.

Response: We have revised figure numbers and figure fonts in the manuscript and Figure files. Please see the attachment.

Reviewer 2 Report

The authors propose interesting

classification of high and low immune subtype

melanoma. Mutations in B2M and PTEN are considered potential therapeutic targets that
343 can improve the immune response.
“High immune subtype” patients can generally obtain a
344 better immune prognosis effect

if TGF-β inhibitors are combined.

The authors validated analysis of tumor

samples on melanoma cell lines.

The study is important to the field.

The minor comments:

Please include more literature support

with clinical studies

on resistance of melanoma samples to

PD1 inhibitors, to TGF1 beta pathway and

discuss the data based on findings of the present study.

Author Response

The authors propose interesting classification of high and low immune subtype melanoma. Mutations in B2M and PTEN are considered potential therapeutic targets that 343 can improve the immune response. “High immune subtype” patients can generally obtain a 344 better immune prognosis effect if TGF-β inhibitors are combined. The authors validated analysis of tumor samples on melanoma cell lines. The study is important to the fields.

The minor comments:

Please include more literature support with clinical studies on resistance of melanoma samples to PD1 inhibitors, to TGF1 beta pathway and discuss the data based on findings of the present study.

Response:

Thank you for your considerate comments which are very useful for us to further improve the manuscript. Please see the following details and the attachment.

-#1. Please include more literature support with clinical studies on resistance of melanoma samples to PD1 inhibitors, to TGF1 beta pathway and discuss the data based on findings of the present study.

Response: I have included and discussed more literature supports in my manuscript. Here are the revised discussion and references. Please see the attachment.

We herein started by considering the tumour microenvironment and tumour cells and built the SING method by immune annotation information as background to explore the immune function of cutaneous melanoma-related subtypes of skin. The SING algorithm is helpful to identify immune functions related to subtypes. The “immune inactivation subtype” had two unique genetic mutations, B2M and PTEN, which meant that they were molecular features of the normal class. Studies have found that B2M is associated with antigen presentation in the tumour immune cycle [49]. B2M gene-encoded β-20 microglobulin (MHC-1) is an important adjunct to major histocompatibility complex class 1, and its absence leads to a decline in MHC-1 expression, affecting antigen presentation and resulting in PD-1 drug resistance [50]. A meta-analysis to systematically review [51] showed that PD-1 inhibitors significantly improved the progression-free survival (PFS), overall survival (OS) and overall response rate (ORR) in patients with advanced melanoma. Anti-PD-1 drugs such as pembrolizumab and nivolumab can achieve long-term survival for patients with metastatic melanoma[52]. PTEN, as a tumour suppressor gene [53], was found only in the normal class, suggesting that PTEN mutations may be a factor affecting immune properties. Therefore, we suggest that they are potential therapeutic targets to improve the immune response. Compared with the “low immune subtype”, “high immune subtype” showed higher immune infiltration and immune activation characteristics and was significantly enriched in TGF-β signalling pathways [54, 55]. TGF-β signalling pathways have been repeatedly reported [54, 56, 57] to be associated with immune regulation, and the use of TGF-β inhibitors may improve the immune response to immunotherapy. In recent years, inhibitors targeting TGF-β pathway have been discovered and investigated by pharmaceutical companies for cancer therapy, and some of them are in clinical trial now, such as Phase I study of GC1008 (fresolimumab), Ph II study of NCT01453361 (Gemogenovatucel-T) and so on [58, 59]. In 2019, Kaczorowski et al. [60] suggested that overexpression of SMAD7 may be a new hallmark inhibitor of TGF-β in melanoma. In 2021, Liu et al. [61] proposed a triple combination therapy with PD-1/PD-L1, BRAF, and MEK inhibitor in stage III-IV melanoma as significantly improving PFS of patients. Based on the risk score between the three immune subtypes, “high immune subtype” showed better prognostic value. Therefore, we speculated that patients with “high immune subtype” biological characteristics can generally obtain a better immune prognosis and that the immune prognosis is improved if combined with TGF-β inhibitors. In the “low immune subtype”, only one significantly mutated gene, NRAS, was found, which was considered a molecular feature of this subtype.

References:
  1. Li J, Gu J: Efficacy and safety of PD-1 inhibitors for treating advanced melanoma: a systematic review and meta-analysis. Immunotherapy 2018, 10(15):1293-1302.
  2. Gellrich FF, Schmitz M, Beissert S, Meier F: Anti-PD-1 and Novel Combinations in the Treatment of Melanoma-An Update. J Clin Med 2020, 9(1).
  3. Huang CY, Chung CL, Hu TH, Chen JJ, Liu PF, Chen CL: Recent progress in TGF-beta inhibitors for cancer therapy. Biomed Pharmacother 2021, 134:111046.
  4. Morris JC, Tan AR, Olencki TE, Shapiro GI, Dezube BJ, Reiss M, Hsu FJ, Berzofsky JA, Lawrence DP: Phase I study of GC1008 (fresolimumab): a human anti-transforming growth factor-beta (TGFbeta) monoclonal antibody in patients with advanced malignant melanoma or renal cell carcinoma. PLoS One 2014, 9(3):e90353.
  5. Kaczorowski M, Biecek P, Donizy P, Pieniazek M, Matkowski R, Halon A: SMAD7 is a novel independent predictor of survival in patients with cutaneous melanoma. Transl Res 2019, 204:72-81.
  6. Liu Y, Zhang XL, Wang GY, Cui XC: Triple Combination Therapy With PD-1/PD-L1, BRAF, and MEK Inhibitor for Stage III-IV Melanoma: A Systematic Review and Meta-Analysis. Front Oncol 2021, 11.

-#2. Extensive editing of English language and style required

Response: Our manuscript has been edited by Elsevier Language Editing Services to enhance readability.     

This manuscript is a resubmission of an earlier submission. The following is a list of the peer review reports and author responses from that submission.

Round 1

Reviewer 1 Report

The studies add to previous attempts to identify features that will predict melanoma patients response to immunotherapy.  It has value in describing a different method of analysis. It is limited in showing just how this information would be used in individual patients and whether it would provide  more information than existing immuno-score  or gene expression approaches

Author Response

- The studies add to previous attempts to identify features that will predict melanoma patients response to immunotherapy.  It has value in describing a different method of analysis. It is limited in showing just how this information would be used in individual patients and whether it would provide  more information than existing immuno-score  or gene expression approaches

Response: Thank you for your positive comment. Please see the attachment.

We calculated the risk score using multivariate Cox regression model according to the literature. Both of the two published signature showed prognostic value when reproducing them in our study (P < 0.001, HR = 1.765, 95% CI: 1.297-2.402; P =0.0667, HR =1.328, 95% CI: 0.9807-1.797). Additionally, we found significant differences in the risk score between the three immune subtypes (P < 0.05). Among them (Supplementary Figure S7), “high immune subtype” had lower risk score, suggesting that “high immune subtype” may have better prognosis, lower tumor purity and active immune-related signaling pathways.

Supplementary Figure S7: There were significant differences between the three immune subtypes in the risk score (P < 0.05).

Reviewer 2 Report

The reviewed manuscript is based on the material from the data bases and as such is not an original research laboratory work.

Basic remark refers “Patients and samples” characteristic. The basic melanoma material came from very heterogenous bases, except TCGA, authors analysed material from cited works in the GEO database. The cited publications 22-25 are based on a very heterogenous clinical material and used different methods for molecular analysis (years of publication 2008-2013).
Cited position 22:  Material consisted of: 40 metastatic melanoma, 42 primary cutaneous melanoma (among them only 16 were the primary melanomas), 4 normal skin.
All these samples gave cited 86 samples in the reviewed manuscript, but not all of them were melanomas.
Cited position 23: Original manuscript analysed 118 patients. Authors chose 22 which one? 
Cited position 24: 44 samples from 38 patients (melanoma stages from I to IV).
Cited position 25: original manuscript analysed: 46 primary melanomas, 12 metastatic melanomas , 16 normal skin. Authors chose 62 which one?

     The material for analysis has not been selected correctly. There are methodological incorrect presumptions (different melanoma stages, other cutaneous tumors analysed together with melanomas). This disqualified the whole done analysis. Statistic is a very useful tool but the most important is an anayzed biological object – melanoma. Therefore, the manuscript should provide relevant information about patient characteristics, biological assays methods etc. The melanoma can not be analysed together with other skin cancers as one group. 

Author Response

- 基本备注是指“患者和样本”特征。基本的黑色素瘤材料来自非常异质的碱基,除了 TCGA,作者分析了 GEO 数据库中引用作品的材料。引用的出版物 22-25 基于非常异质的临床材料,并使用不同的方法进行分子分析(出版年份 2008-2013)。
所有这些样本在审查的手稿中引用了 86 个样本,但并非所有样本都是黑色素瘤。

Response: Thank you for your considerate comments which is very useful for us to further improve the manuscript. Please see the following details and the attachment.

- Cited position 22:  Material consisted of : 40 metastatic melanoma, 42 primary cutaneous melanoma (among them only 16 were the primary melanomas), 4 normal skin.

Response: We have carefully checked each external cohort enrolled in our study. To be specific, GSE7553 included 86 samples where 82 samples are tumors and 4 samples are normal. Thus, we chose 82 melanoma samples (40 metastatic melanoma, 42 primary melanoma) for the purpose of the study.

- Cited position 23: Original manuscript analysed 118 patients. Authors chose 22 which one? 

Response: We apologize for this confusion. Actually, the validation cohort we used here is derived from GSE23376 which included 22 melanoma samples. We checked carefully and did not find a proper literature to cite for this cohort, we therefore cited another literature published by the same author and also provided a website link to navigate to the validation cohort (https://www.ncbi.nlm.nih.gov/geo/query/acc.cgi?acc=GSE23376). 

- Cited position 24: 44 samples from 38 patients (melanoma stages from I to IV).

Response: This literature corresponding to GSE19234 that included 44 tumor samples derived from 38 patients. We used all the samples of this cohort for the purpose of our study.

- Cited position 25: original manuscript analysed: 46 primary melanomas, 12 metastatic melanomas , 16 normal skin. Authors chose 62 which one?

Response: We believe the literature corresponding to GSE15605 which included 74 samples (58 tumor samples and 16 normal samples), we choose 58 melanoma samples from this cohort, including 46 primary melanomas and 12 metastatic melanomas for the purpose of the study.

- The material for analysis has not been selected correctly. There are methodological incorrect presumptions (different melanoma stages, other cutaneous tumors analysed together with melanomas). This disqualified the whole done analysis. Statistic is a very useful tool but the most important is an anayzed biological object – melanoma. Therefore, the manuscript should provide relevant information about patient characteristics, biological assays methods etc. The melanoma cannot be analysed together with other skin cancers as one group. 

Response: We sincerely thank the reviewer for pointing out such deficiency. The valuable comment raised by the reviewer will definitely help us to improve the methodology of this study. According to the comment, we added the additional 12 metastatic melanomas from GSE15605 to our validation part. Therefore, a total of 206 melanoma samples were used and re-analysed for validation in the revised manuscript. The results were quite consistent with previous version, which suggested that specific immune subtypes were in skin melanoma and their immune responses varied according to subtype characteristics. Additionally, to enhance readability of our manuscript for the enrolled melanoma cohorts, we provided a table that summarized the datasets that were used in this study. Please kindly see supplementary Table S1 for more details. Again, thanks for this pertinent and valuable comments.

Method

The discovery cohort used in this study was from The Cancer Genome Atlas (TCGA) under the SKCM archive with a total of 327 tumor samples; gene expression, mutation, DNA methylation, and clinical data were retrieved then. External validation cohort were obtained from the GEO database (GSE7553 (22) [n = 82], GSE23376 (23) [n = 22], GSE19234 (24) [n = 44], GSE15605 (25) [n = 58]). 206 tumor samples (Supplementary Table S1) were finally merged based on elimination of batch effects. Immune annotation information was obtained from the MSigDB database.

Result

In this study, four independent cohorts were integrated as a verification set for supervised analysis (206 tumor samples). First, the existence of immune class (Cluster1) and normal class (Cluster2) were verified, the immune class had higher immune signatures (Supplementary Figure S6). To further validate immune subtypes based on tumor microenvironment, three subtypes of the TCGA cohort were analyzed for pairwise differential expression, and the corresponding differential expression genes were selected as the specific signatures of the three SKCM immune subtypes. Among them, a 737-gene signature was identified for “immune inactivation subtypes”, a 525-gene signature for “low immune subtype” and an 839-gene signature for “high immune subtype”. Using the above characterization gene signatures for supervised consensus clustering (Figure 5a), three subtypes of validation cohort were reproduced: “high immune subtype” (98 samples), “low immune subtype” (58 samples), and “immune inactivation subtypes” (50 samples). Among three subtypes, significant differences were observed in cell cytotoxicity and cytolysis, immune score and stromal score (Figure 5b-d, P < 0.05). These suggested that specific immune subtypes were in skin melanoma and their immune responses varied according to subtype characteristics.

补充表 S1。 研究中包含的四个黑色素瘤数据集的总结。 

数据集

档案

平台

数据类型

SKCM数量

患者人数

参考

SKCM

TCGA

Illumina HiSeq 2000 RNA 测序

RNA序列

327

327

是的

GSE7553

地球观测站

Affymetrix 人类基因组 U133 Plus 2.0 阵列 [HG-U133_Plus_2]

微阵列

82

82

是的

GSE23376

地球观测站

Affymetrix 人类基因组 U133 Plus 2.0 阵列 [HG-U133_Plus_2]

微阵列

22

22

/

GSE19234

地球观测站

Affymetrix 人类基因组 U133 Plus 2.0 阵列 [HG-U133_Plus_2]

微阵列

44

38

是的

GSE15605

地球观测站

Affymetrix 人类基因组 U133 Plus 2.0 阵列 [HG-U133_Plus_2]

微阵列

58

58

是的

Reviewer 3 Report

Regarding the methodology followed for the grouping of the cases under study, it has a good approach

The authors have reached a good approximation in the best of treatments considering microenvironment based on immune annotation to contribute to the mining of skin cutaneous melanoma immune function-related subtypes.

Just note that the reference 2 is missing 

Finally, congratulate the authors for their work.

Author Response

- Regarding the methodology followed for the grouping of the cases under study, it has a good approach.The authors have reached a good approximation in the best of treatments considering microenvironment based on immune annotation to contribute to the mining of skin cutaneous melanoma immune function-related subtypes.

Response 1: Thank you for your positive comment.

- Just note that the reference 2 is missing.

Response 2: I have added the reference in my manuscript. Here is the reference information.  Please see the attachment.

2. Gendoo DM, Ratanasirigulchai N, Schroder MS, Pare L, Parker JS, Prat A, Haibe-Kains B: Genefu: an R/Bioconductor package for computation of gene expression-based signatures in breast cancer.

Reviewer 4 Report

This work classified the skin cutaneous melanoma samples based on the immune function. Three subtypes were proposed according to the sample-specific networks and the tumor microenvironment. The study is of significant, and my only concern is the small number of the research samples which might affect the conclusion.

Round 2

Reviewer 2 Report

I appreciate done correction about the analyzed melanoma material. The available material has not been selected concerning the melanoma biology (all stages that have different biology are analyzed together) and authors did not add any novelity to the melanoma biology. This analysis could be a presumption to a real experimental work that could give new values to the melanoma cure. In my opinion without this experimental part the manuscript is not ready to be published as the original scientific work.

Reviewer 3 Report

The authors have reflected what was indicated in my review.